# FUNU: Boosting machine unlearning efficiency by filtering unnecessary unlearning

## Abstract

Machine unlearning is an emerging field that selectively removes specific data samples from a trained model. This capability is crucial for addressing privacy concerns, complying with data protection regulations, and correcting errors or biases introduced by certain data. Unlike traditional machine learning, where models are typically static once trained, machine unlearning facilitates dynamic updates that enable the model to "forget" information without requiring complete retraining from scratch. There are various machine unlearning methods, some of which are more time-efficient when data removal requests are fewer.

To decrease the execution time of such machine unlearning methods, we aim to reduce the size of data removal requests based on the fundamental assumption that the removal of certain data would not result in a distinguishable retrained model. We first propose the concept of unnecessary unlearning, which indicates that the model would not alter noticeably after removing some data points. Subsequently, we review existing solutions that can be used to solve our problem. We highlight their limitations in adaptability to different unlearning scenarios and their reliance on manually selected parameters. We consequently put forward FUNU, a method to identify data points that lead to unnecessary unlearning. FUNU circumvents the limitations of existing solutions. The idea is to discover data points within the removal requests that have similar neighbors in the remaining dataset. We utilize a reference model to set parameters for finding neighbors, inspired from the area of model memorization. We provide a theoretical analysis of the privacy guarantee offered by FUNU and conduct extensive experiments to validate its efficacy.

## CCS Concepts

• **Security and privacy**;

## Keywords

Machine unlearning, Data selection, Data prototype

**ACM Reference Format:**
Anonymous authors. 2018. FUNU: Boosting machine unlearning efficiency by filtering unnecessary unlearning. In *Proceedings of Make sure to enter the correct conference title from your rights confirmation emai (Conference acronym 'XX)*. ACM, New York, NY, USA, 12 pages. https://doi.org/XXXXXXX.XXXXXXX

## 1 Introduction

To enforce privacy regulations that protect individuals' right to be forgotten [3, 12], machine unlearning [6] has been proposed to remove specific data samples from a trained model. It can also be used to remove harmful data from the model, thereby mitigating potential risks [7, 39].

While most methods aim to approximately unlearn a model without retraining it from scratch, some of them would take longer time to unlearn as the size of data removal requests grows[2, 11, 23, 35, 42]. For instance, SISA (Sharded, Isolated, Sliced, and Aggregated) [2] is one of these methods. It partitions the entire dataset into slices and trains sub-models on these slices. When data removal requests are received, only the sub-models of the affected slices are retrained, thereby avoiding a complete retraining. However, as the number of removal requests increases, more slices are affected, potentially leading to longer unlearning time.

However, we argue that many removal requests are unnecessary, as there is redundant information among similar samples. In other words, if a sample has similar neighbors in the remaining dataset, unlearning it would not produce a model distinguishable from the model before unlearning. Figure 1 illustrates an example.

EXAMPLE 1.1. *Consider a recommender system trained on user data. Suppose there are ten users with similar purchase records as Amy, whereas for Bob, no other user has similar purchase records. As such, removing Amy's data would become unnecessary because it would have a smaller impact on the model than removing Bob's data, and other users with similar records to Amy would contribute to the model as if Amy's data still existed.*

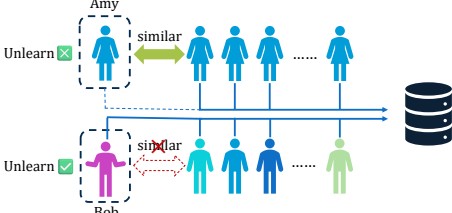

**Figure 1: Example of unnecessary unlearning**

In this work, we aim to detect those samples in data removal requests that would result in unnecessary unlearning. Existing works [9, 16, 40] on data prototype discovery, as reviewed in Section 2.2.1, can be applied to our problem. However, these approaches are not designed for machine unlearning settings and have two primary limitations. First, they lack the adaptability to different types of unlearning scenarios, such as random removal and class removal. Second, their performance heavily depends on the internal parameters, and manual tuning is required to select these parameters.

To overcome these limitations, we further propose FUNU (Filtering UNnecess-ary Unlearning), which addresses these limitations in

two aspects. First, FUNU adopts a distance measure between the remaining dataset and the removal requests to find samples that result in unnecessary unlearning, so it can adapt to different unlearning scenarios. Second, FUNU trains a reference model in one epoch and uses it for parameter tuning, especially the similarity thresholds, without manual intervention. To guarantee that FUNU satisfies the definition of approximate machine unlearning, we also analyze the theoretical bound of the output distance between the model generated by FUNU and a retrained model. Experimental results demonstrate that FUNU can enhance the efficiency of those machine unlearning methods whose cost is proportional to the size of data removal requests while meeting the same commitment of the right to be forgotten.

Besides, for other machine unlearning methods whose cost does not proportionally increase [17, 19, 21, 36], FUNU can also benefit them. For example, many such methods are constrained by a deletion capability [36] that limits the total number of removal requests they can accommodate under a privacy guarantee. Since FUNU reduces the number of removal requests, the validity of such methods is extended. In addition, FUNU also aligns with the robustness of the model, as the training process is intended to produce a model that performs stably when the dataset is slightly perturbed [43]. To conclude, our main contributions are as follows.

- We are the first to point out the unnecessary unlearning phenomenon and propose to enhance the efficiency of machine unlearning methods by exploiting it.
- We put forward FUNU to filter samples that lead to unnecessary unlearning. FUNU avoids the limitations of existing solutions. It adapts to different unlearning scenarios and does not require prior knowledge to choose parameters.
- We conduct a theoretical analysis to guarantee that the model generated by FUNU is close to the retrained model. Moreover, we perform extensive experiments to demonstrate that it can indeed improve the efficiency of machine unlearning methods.

The organization of this paper is as follows. Section 2 introduces preliminaries, including problem setting and limitations in existing solutions. Section 3 proposes our method FUNU as well as its privacy guarantee. Section 4 presents empirical evidence to validate FUNU. Section 5 introduces related work briefly and Section 6 finally concludes this paper.

## 2 Problem setting and existing solutions

In this section, we formulate the concept "unnecessary unlearning", and then introduce the problem we aim to address in this paper. We also present a few baseline solutions adapted from existing works and show their limitations.

### 2.1 Problem setting

In the context of machine unlearning, let $A$ denote the training process, so $A(D)$ trains a model on dataset $D$. we start with an original model $M_o$ trained on the complete dataset $D_o$, i.e., $M_o = A(D_o)$. Subsequently, data removal requests $D_u$, also referred to as the unlearning dataset, are received. The model trained on remaining dataset $D_r = D_o \backslash D_u$ is denoted as $M_r = A(D_r)$.

To describe the similarity between models, we use readout functions [17–19]. Readout functions $f(M)$ indicate the information that can be extracted given a model $M$. Common readout functions include model output, accuracy on the model prediction, etc. $Dist(\cdot)$ is the distance measurement to quantify the distance between the outputs of readout functions for different models. It can be measured using metrics such as KL-divergence, norm, etc. With these components, the definition of unnecessary unlearning is as follows.

*Definition 2.1 (Unnecessary Unlearning).* Given the remaining dataset $D_r$, the data removal requests $D_u$, and the training process $A$, the unlearning process over $D_u$ is $\epsilon$-unnecessary unlearning if

$$Dist(f(A(D_r)), f(A(D_r \cup D_u))) \le \epsilon$$

In the above definition, $Dist(f(A(D_r)), f(A(D_r \cup D_u)))$ aligns with the definition of previous unlearning works [17–19]. However, these works employ different choices for $Dist(\cdot)$ and the readout functions. The underlying idea of this definition is that, an unlearning process which would not produce a model distinguishable (as measured by $\epsilon$) from the original one, is considered unnecessary unlearning.

Now we formulate the problem in this paper as follows. Given a remaining dataset $D_r$, an unlearning dataset $D_u$, and the training process $A$, how can we **find a subset $D_u^+ \in D_u$, so that unlearning over $D_u^+$ would still guarantee $\epsilon$-unnecessary unlearning?**

After filtering $D_u^+$ from $D_u$, the unlearning process would continue to be performed on the remaining removal requests $D_u^- = D_u \backslash D_u^+$. Since $|D_u^-|$ is smaller than $|D_u|$, unlearning $D_u^-$ would consume less time and resources.

### 2.2 Solutions by data prototype discovery

A highly relevant problem to unnecessary unlearning is data prototype discovery, which finds typical samples that best represent the whole dataset [8]. The discovered prototypes can be regarded as the summary of the dataset to enhance training efficiency or to explain model behavior [45]. Table 1 summarizes existing works in data prototype discovery. They can be categorized into five types.

(1) For methods in [37], [10] and the adv indicator proposed in [8], Membership Inference Attack (MIA) or adversarial attack is required to decide the typicality of data points. However, MIA and adversarial attacks need to train additional models, which could incur costs comparable to retraining a model, contradicting the objective of machine unlearning and unnecessary unlearning.

(2) For methods in [13], [14], priv, ret and agr[8], they select prototypes by testing on models trained without the selected data, which inherently includes a retraining process in their methodology. As such, these methods also conflict with the objective of machine unlearning.

(3) For methods in [29], [31], and [45], they attempt to modify model architecture or alter the training process, which is impractical in machine unlearning where the model has already completed training.

(4) Method in [26] involves searching prototypes by an NP problem, which would lead to high computation costs.

**Table 1: Existing data prototype discovery methods and their limitations in efficiency**

| Methods | Main idea | Limitations |
|---|---|---|
| [37] [10] adv[8] | Perform MIA or adversarial attack to see the typicality of a given sample | Require MIA or adversarial attack |
| [13], [14], ret,priv, agr[8] | Compare model stability trained with or without a certain sample | Require retraining or training |
| [29],[31], [45] | Track training process or add additional network | Need to modify the model architecture or training process |
| [26] | Use a metric to select typical samples | Need to solve an NP problem |
| Existing solutions [8, 16, 34] | see Section 2.2.1 | - |

(5) **clustering[5, 8], confidence[8] and curvature[16, 34]** are more efficient data prototype discovery solutions that can be adapted to our problem as detailed below.

*2.2.1 Adaptation to unnecessary unlearning problem.* To adapt to our problem, these solutions share a common three-step process. Since they only differ in the first step, we will elaborate on this step in greater detail.

**Step 1.** Calculate a score $s(x)$ for each sample $x$. The score reflects whether a sample has many similar neighbors in the entire dataset, consistent with the indications of other data prototype discovery methods mentioned previously. In our design, samples with more similar neighbors tend to score lower. The ways to calculate scores for different methods are as follows.

- Clustering. Inspired by [5, 8], we employ a combination of dimensionality reduction and clustering techniques. We apply t-SNE[38] on the pixel space (for dataset MNIST in our experiment) and ResNet-generated feature space (for datasets CIFAR-10 and CIFAR-100) to project the datasets into two dimensions and cluster with HDBSCAN. $s(x)$ for each sample $x$ is its according outlier score [4] in the clustering process.

- Confidence[8]. Model confidence is the output of the final fully connected layer, indicating the probability of the sample belonging to each class. We use softmax to normalize the model output. To adjust the direction of the score, we assign $1 - softmax(c(x))$ as the confidence score of $x$, where $c(x)$ is sample $x$'s belonging class and $softmax(i)$ is the model output after softmax on class $i$.

- Curvature[16, 34]. The curvature of the network loss around a data point indicates the model's memorization of it. Less rare samples tend to have low curvature. In the original paper [16], the authors averaged curvatures over many epochs to calculate the final score. However, curvature calculation is timing-consuming (in our experiment, it takes more than

one hundred seconds to calculate curvature for one epoch). To maintain competitiveness with other methods, we calculate the curvature only at the second epoch as our $s(x)$. We choose curvature at this epoch because at this stage, the model does not learn too much about the dataset and thus the curvature could reflect how the model reacts to a sample more accurately. Otherwise, if the model were well-trained, it would fit most samples, resulting in uniformly low curvature and it is hard to see the difference.

**Step 2.** Rank the scores $s(x)$ in ascending order. A lower score indicates the sample has more similar neighbors or is more typical within the entire dataset.

**Step 3.** Select $D_u^+ = \{x | s(x) \leq \theta \text{ and } x \in D_u\}$. As the parameter $\theta$ requires manual selection, it could lead to extensive testing for different parameter choices. In Section 4.2, we select parameters based on the data distribution and show that the results are sensitive to the choice of parameters.

## 2.3 Limitations of existing solutions

Even though the above solutions are reasonably efficient as baselines, they have the following limitations.

**Limitation 1.** These methods fail to adapt to different types of unlearning scenarios. For example, they can not automatically adjust in random removal and class removal scenarios. The former is to randomly remove samples regardless of their associated classes, and the latter is to remove all samples of a particular class. Both scenarios are common in practical unlearning applications.

In the class removal scenario, as the removed samples belong to the same class, intuitively the proportion of $D_u^+$ should be smaller than in random removal, because neighbors of samples in the same class are likely remain in this class and are thus also subject to unlearning. Existing works fail to identify this difference and thus would leave out samples supposed to be unlearned. Figure 2 illustrates this. Existing works select $D_u^+$ by evaluating the contribution of each sample in the entire dataset. Specifically, they tend to pick samples that have many similar neighbors in the whole dataset to be $D_u^+$. In this case, as the relationship between the given sample and the entire dataset is fixed, whether a sample would be classified into $D_u^+$ is also fixed. Therefore, the proportion of $D_u^+$ remains fixed under different unlearning scenarios.

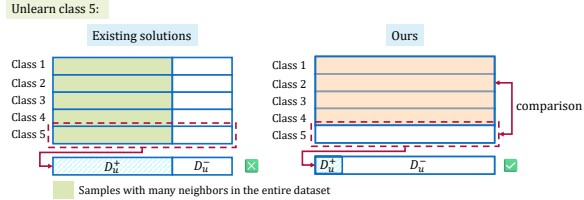

**Figure 2: Existing solutions and our method in class removal scenario. As for existing solutions, they tend to select samples with many neighbors in the entire dataset as $D_u^+$. As the proportion of such samples is fixed throughout the dataset, thus the proportion of $D_u^+$ would be consistent with that and be high. However, in our design, as we compare the removal requests with the remaining dataset, thus the proportion of $D_u^+$ would be low.**

We believe the root cause is the difference of motivations. Existing works evaluate the **absolute** contribution of each sample in the **entire** dataset, and attempt to filter those that contribute less than the remaining removal samples so that the model trained on the latter would resemble the original model $M_o$. In our problem, however, we hope to find and filter those samples that cause a model to resemble the retrained model $M_r$, so we should focus on the **relative** contribution to the **remaining** dataset.

**Limitation 2.** These methods require manual parameter selection. Our experiments in Section 4.2.4 show that they are sensitive to parameters. For instance, a 0.01 decrease in the parameter will reduce the proportion of remaining removal requests by approximately 20% when applying the method "curvature". which makes it challenging to derive or tune the parameter in practice.

# 3 FUNU: an unnecessary unlearning filtering method

FUNU aims to select a subset of unlearning dataset $D_u$ to form $D_u^+$ such that the samples in $D_u^+$ have similar neighbors in $D_r$. The contribution of these samples would be replaced by neighbors in $D_r$, rendering them less significant to the retrained model. Consequently, unlearning operations can bypass these samples.

In detail, FUNU has three steps, as illustrated in Figure 3. First, calculate the distance matrix based on features generated by the original trained model $M_o$ (Section 3.1). Second, use a reference model $M_{ref}$ to establish the similarity condition (Section 3.2). If a sample $x$ and a dataset $D$ satisfy this condition, then the sample $x$ would be considered to have sufficient similar neighbors in $D$. Third, we compare samples in $D_u$ with $D_r$ using the similarity condition established in the previous step and select $D_u^+$ (Section 3.3).

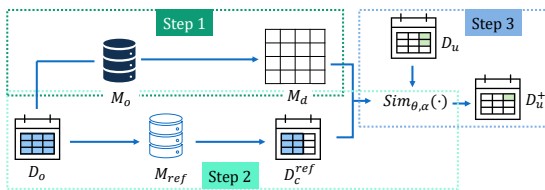

**Figure 3: The procedure of FUNU**

## 3.1 Calculate distance matrix

The distance matrix, denoted as $M^d$, is a matrix whose dimension equals the size of the original dataset, i.e., $M^d \in \mathbb{R}^{N \times N}$, where $N = |D_o|$. Each cell of $M^d$ represents a distance between two samples, such that $M^d_{i,j} = sample\_dist(x_i, x_j)$, where $x_i, x_j \in D_o$.

We now introduce the function above $sample\_dist$ in detail. We first use the original model $M_o$ to generate the feature representation of a given sample $x$. Specifically, we use the model output before the last Fully Connected (FC) layer as the feature representation of the given sample, denoted as $M_f(x)$. Next, for each sample pair in $D_o$, we calculate their cosine distance. Combining these two steps, we have $sample\_dist(x_i, x_j) = cos(M_f(x_i), M_f(x_j))$, which constructs the distance matrix. It is important to note that the larger the cosine distance, the more similar the two samples are. The distance matrix is further utilized in the third step.

## 3.2 Establish similarity condition

The similarity condition $Sim(x, D)$ determines whether a sample $x$ has a sufficient number of similarity neighbors in dataset $D$, such that the contribution of $x$ to a model trained on $D$ could be considered negligible.

We formulate the similarity condition as follows. $Sim_{\theta,\alpha}(x, D) = True$ if there are more than $\alpha$ samples $y_1, ..., y_{\alpha*}(\alpha* \geq \alpha, y_i \in D)$ that satisfies $sample\_dist(x, y_i) \geq \theta$. If there are enough samples in $D$ considered similar to $x$, $Sim_{\theta,\alpha}(x, D)$ will return true.

The key problem in establishing the similarity condition is to determine the parameters, $\alpha$ and $\theta$. Instead of manually choosing parameters, we use a reference model to find similar samples first and derive the statistics of those samples as the parameters.

Our reference model $M_{ref}$ is acquired as follows. We initialize a model with the same structure as the original model and train it on $D_o$ for one epoch. The resulting model is designated as $M_{ref}$. The choice of $M_{ref}$ is motivated by two key considerations.

First, regarding the efficacy of $M_{ref}$ in addressing our problem, We would like to use it to identify similar samples. Since $M_{ref}$ iterates over the whole dataset for one epoch, it has seen each data sample only once. In this case, we infer the samples correctly predicted by $M_{ref}$ have sufficient similar neighbors in the entire dataset, enabling the model to make correct predictions. This inference is consistent with the findings in model memorization area [1, 40], which suggest that models tend to learn patterns at the early training stage. $M_{ref}$ is a model at an early stage of training (one epoch). The data samples being predicted correctly could be recognized as contributing to pattern learning, and thus share common characteristics. Second, considering efficiency, our $M_{ref}$ is straightforward to implement. It does not require additional datasets for training and only trains for one epoch, thereby conserving computational resources and time.

Samples correctly predicted by $D_o$ within each class are considered similar. We denote $D_c^{ref}$ as the samples in class $c$ that are correctly predicted by $M_{ref}$. Then we use the statistics of $D_i^{ref}$ across different classes to be parameters for $Sim_{\theta,\alpha}$. Specifically,

$$\theta_c = avg(sample\_dist(x_i, x_j)), x_i, x_j \in D_c^{ref} \quad (1)$$
$$\theta = avg(\theta_c), \ c \in C \quad (2)$$

where $avg(\cdot)$ is the average operation and $C$ is the set of classes. As for the parameter $\alpha$, we first count $\alpha_c$ within each class $c$, which is the count of the sample pairs whose $sample\_dist$ is above the given $\theta$. Then we average $\alpha_c$ across different classes to have the final $\alpha$. This process is formalized below where $|\cdot|$ indicates the size of a given set:

$$\alpha_c = |\{(x_i, x_j)|sample\_dist(x_i, x_j) \geq \theta\}|, x_i, y_i \in D_c^{ref} \quad (3)$$
$$\alpha = avg(\alpha_c), \ c \in C \quad (4)$$

## 3.3 Filter removal requests

We utilize the distance matrix $M^d$ and the parameterized similar condition $Sim_{\theta,\alpha}(x, D)$ to filter samples in $D_u$ that have more than $\alpha$ similar neighbors in $D_r^{c(x)}$, where $D_r^{c(x)}$ is the samples in $D_r$ the belong to the same class $c(x)$ as sample $x$. This process is formalized as follows: $D_u^+ = \{x|x \in D_u \text{ and } Sim_{\theta,\alpha}(x, D_r^{c(x)}) = True\}$.

Specifically, we select a sub-matrix of $M^d$, which contains distances between samples in $D_u$ and $D_r$, i.e., the sub-matrix is shaped $|D_u| \times |D_r|$. Then we perform filtering on this sub-matrix using the formalization we mentioned beforehand, comparing the distance of sample pairs with the similarity condition. Selected $D_u^+$ will be removed from $D_u$ and the remaining samples, denoted as $D_u^-$, are those that require unlearning.

It is noteworthy that this step has to be executed whenever data removal requests are received, as the remaining dataset $D_r$ changes each time. As demonstrated in our experiment (Section 4.2.2), this incurs additional time costs compared to existing solutions, but it is necessary so that our method can adapt to various unlearning scenarios.

## 3.4 Privacy guarantee

We claim that the $D_u^+$ selected by FUNU satisfies $\epsilon$-unnecessary unlearning in Section 2.1. To specify the definition of unnecessary unlearning, we choose KL-divergence as the distance measurement. As illustrated in Section 2.1, the difference between $M_u$ and $M_r$ is that $M_u$ is trained on $D_u^+$ while $M_r$ is not. Consequently, we use the model outputs on $D_u^+$ as the readout function to better illustrate the differences between the two models.

It is important to note that we compare similarities between samples using the features that are generated just before the last FC layer, Thus the relationship between features and outputs can be regarded as linear.

Besides, in previous machine unlearning studies that aim to bound the distance between the retrained model and unlearned model, it is a common assumption that model outputs on samples are independent of each other [21, 36, 41], i.e., if the training dataset of a model contains a certain sample, then the changes of other samples (removal or addition) would not affect the model output on that sample. Following this convention, we assume that the features produced by $M_o$ are identical to $M_r$ and $M_u$ as the training dataset of $M_o$ covers that of $M_r$ and $M_u$.

In addition, Given that both $M_u$ and $M_r$ are trained on $D_r$, we assume that $\|log(M_r(x)) - log(M_u(x))\| \leq \delta$ on $D_r$. This implies that the logarithms of the outputs of $M_u$ and $M_r$ on samples in $D_r$ are bounded by a small value $\delta$. Since we have hypothesized the same features for $M_r$ and $M_u$ before the FC layer, the logarithm of the model output equals the logarithm of the final FC layer output. $\delta$ can be regarded as an estimation of the possible output difference between two models trained on the same dataset. Ideally, the expectation of $\delta$ is zero.

Based on these prerequisites, we have the following theorem.

THEOREM 3.1. *Suppose that for model $M_r$ and $M_u$, the logarithms of final FC layer output, denoted as $log(M_r)$ and $log(M_u)$, are $\lambda_1$-Lipschitz and $\lambda_2$-Lipschitz, and that $\|log(M_r(x)) - log(M_u(x))\| \leq \delta$ on $D_r$, then*

$$KL_{D_u^+}(p_u \| p_r) \leq \epsilon, \epsilon = n[(\lambda_1 + \lambda_2)(\sqrt{2 - 2\theta}) + \delta]$$

*where $p_u$ is the output distribution of $M_u$, and $p_r$ is that of $M_r$. $n$ is the size of $D_u^+$.*

The proof of the theorem is in appendix A.

**Table 2: Datasets and models**

| Datasets | Length | Dimensions | Models |
|---|---|---|---|
| MNIST[28] | 60,000 | 28×28 | 2-layer-CNN |
| CIFAR-10[27] | 50,000 | 32×32×3 | ResNet-18[22] |
| CIFAR-100[27] | 50,000 | 32×32×3 | ResNet-18 |

## 4 Experiment

We conduct two sets of experiments. The first experiment is to evaluate our method FUNU with other existing solutions from the aspects of efficiency, adaptivity, and model privacy. The second is to apply FUNU to an unlearning method SISA [2], demonstrating that FUNU can improve efficiency while preserving model similarity. Our implementation is available at https://anonymous.4open.science/r/unnecessary_unlearning-BEE3.

## 4.1 General experiment setting

We followingly introduce datasets, models, and metrics used throughout the experiments. The particular settings for the two experiments are specified in their respective sections.

**Datasets and models.** We train models on the datasets as listed in Table 2. We use three datasets: MNIST, CIFAR-10, and CIFAR-100. We employ a 2-layer convolutional neural network (2-layer-CNN) for MNIST and a ResNet-18 model for CIFAR-10 as well as CIFAR-100. We sample 90% of the complete dataset separately for training our own model and the shadow model in MIA. When training the ResNet-18 model, we optimize the pre-trained model using Stochastic Gradient Descent with a learning rate of 1e-2 for CIFAR-10 and 2e-4 for CIFAR-100, for ten epochs.

**Metrics.** We use metrics from three aspects: (1) Time cost. We record the time required to execute different algorithms to illustrate their efficiency. (2) Reduction in the data removal requests. This metric assesses the extent to which the methods can reduce data removal requests. It is quantified by the proportion of remaining removal requests in the original requests, denoted as $P^- = \frac{|D_u^-|}{|D_u|}$. A larger $P^-$ indicates that fewer removal requests have been reduced. (3) Model privacy. Model privacy is measured by the similarity between the model $M_u$ trained on $D_u^+ \cup D_r$ and the retrained model $M_r$ trained on $D_r$. If the two models are similar, then we could conclude that $M_u$ maintains a level of privacy protection comparable to $M_r$. We thereby compare their performance across several metrics [44], as detailed below, to illustrate their similarity.

- Model accuracy (acc.). We compare the accuracy of $M_u$ and $M_r$ on the removal $D_u$, remaining dataset $D_r$, and test dataset $D_t$. This comprehensive comparison provides a detailed description of the models' performance.
- Accuracy and F1 of MIA [30] on the original data removal requests $D_u$. MIA aims to determine whether a specific data point was part of the training dataset used to build a machine-learning model. By analyzing the model's outputs, the attacker can infer the presence or absence of particular samples. We aim for $M_u$ to achieve similar performance to $M_r$, ensuring that both models react similarly to MIA.

All experiments are performed on NVIDIA GeForce RTX 3090 with CUDA Version 12.2 and implemented with Python 3.8.19.

## 4.2 Evaluation on FUNU

In this section, we compare FUNU with existing solutions from the perspectives of time cost, adaptivity to different unlearning scenarios, and model privacy.

### 4.2.1 Experiment setting.
We provide detailed information regarding the experimental setup as follows.

**Parameter.** The existing solutions necessitate the manual setting of the parameter $\theta$ which serves as a threshold when selecting $D_u^+$. In our experiments, with the calculated score $s(x)$ for each sample $x$ using the existing solutions, we select three values for $\theta$ based on score distribution: $max(avg(s) - std(s), 1e-3)$, $avg(s)$, and $avg(s) + std(s)$, where $avg(s)$ and $std(s)$ represent the average value and standard deviation of score $s(x)$ for all samples, respectively. We choose $max(avg(s) - std(s), 1e-3)$ here because in some cases $avg(s) - std(s) \leq 0$, thereby we bound the value with 1e-3.

**Unlearning scenarios.** We consider two unlearning scenarios. The first is random removal, where we randomly select 30, 50, and 100 samples from the original dataset $D_o$ as the removal requests $D_u$. The second is class removal, where we randomly choose one class and remove half of its samples as $D_u$.

**Baseline.** To demonstrate model privacy, we compare our method with another method, namely Certified Removal (CR) [21], to illustrate how close the models generated with our method as well as existing solutions are to the retrained model. It is important to note that CR is not a method for reducing removal requests. Rather, it is a typical unlearning method that directly updates the model parameter and unlearns the given removal requests [44]. Due to memory constraints when running CR for the ResNet-18 model, we apply the method described in [32] to reduce the model layers to be updated.

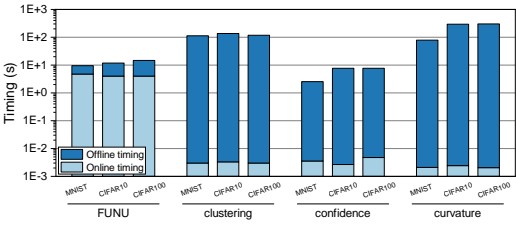

**Figure 4: Timing of FUNU and existing solutions**

### 4.2.2 Time cost.
We calculate the average execution time of these methods under varying numbers of data removal requests in the random removal scenario and present the results in Figure 4.

We divide FUNU and the existing solutions into two stages: the offline and the online stages. For all methods, the offline stage is the first two steps, while the online stage is the third step.

In Figure 4, the offline execution time of FUNU is less than that of the existing solutions, whereas the online execution time exceeds the existing solutions. This discrepancy is expected, as during the online stage, the FUNU compares samples in $D_u$ and $D_r$, which involves the operation of separating the distance matrix and filter value with similar conditions, while existing solutions simply select samples whose scores are smaller than the given parameters.

Nevertheless, the total execution time of FUNU remains less than that of two of the existing solutions, clustering and curvature.

Besides, the absolute online timing of FUNU is around four seconds, which is still acceptable.

### 4.2.3 Adaptivity to unlearning scenarios.
We calculate $P^-$ under both random removal and class removal scenarios. For existing solutions, we first calculate their average $P^-$ across different parameter choices. Subsequently, we calculate the average $P^-$ for FUNU and the existing solutions over varying numbers of removal requests. The results are presented in Figure 5.

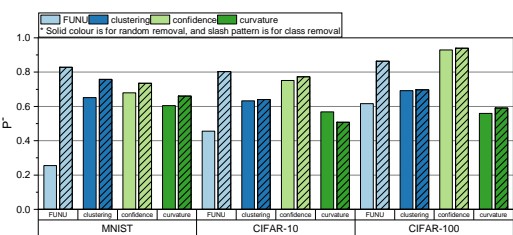

**Figure 5: $P^-$ of different methods**

In Figure 5, in random removal, the average $P^-$ of FUNU is 0.4422, while that of clustering is 0.6586, of confidence, 0.7865, and of curvature, 0.5772. The results indicate that our method, FUNU, is more effective in reducing deletion requests for the random removal scenario compared to existing solutions.

Besides, our method demonstrates adaptability to the class removal scenario, addressing the first limitation mentioned in Section 2.3. When the removal requests are of the same class, $P^-$ of FUNU increases, indicating that FUNU finds more samples are needed to unlearn as they have fewer similar neighbors in the remaining dataset. In contrast, the existing solutions exhibit no obvious variation between the two different unlearning settings, which is due to their neglect of the relationship between $D_u$ and $D_r$, as we have illustrated in Section 2.3.

### 4.2.4 Model privacy.
We use performance similarity between $M_u$ and $M_r$ to illustrate model privacy. Table 3 lists the performance difference. To show the similarity between FUNU (denoted as "ours" in table) and retrained model $M_r$ (denoted as "ret." in table), we use another machine unlearning method, CR [21], as a baseline. Except for the underlined values, our method is closer to the retrained model compared to the baseline. The mean accuracy difference between FUNU and $M_r$ over the three datasets is 0.0134, whereas for CR, it is 0.0535. Similarly, the mean difference between FUNU's corresponding F1 of MIA and $M_r$ is 0.0407, while for CR, it is 0.1114. These results indicate that the model generated by FUNU is more similar to $M_r$, thereby demonstrating superior model privacy.

The performance of models generated using existing solutions is shown in Table 4, along with the chosen parameter $\theta$. For existing solutions, their performance and $P^-$ are sensitive to parameters. Taking curvature as an example, it corresponds to an average rate of change between $P^-$ and the parameter of 20.3589, i.e., a 0.01 decrease in $\theta$ will reduce $P^-$ by approximately 20%. The average rate of change for confidence is 5.0143, and for clustering, it is 1.7352. This parameter sensitivity can lead to significant instability and inconvenience when applying existing solutions.

**Table 3: Model performance under random removal**

| Dataset | Method | Acc. of MIA | F1 of MIA | Acc. on $D_r$ | Acc. on $D_u$ | Acc. on $D_t$ |
|---|---|---|---|---|---|---|
| MNIST | ret. | 0.5124 | 0.4284 | 0.9567 | 0.9722 | 0.9602 |
| | ours | 0.5374 | 0.4732 | 0.9570 | 0.9703 | 0.9602 |
| | CR | 0.5194 | 0.6133 | 0.9564 | 0.9967 | 0.9587 |
| CIFAR-10 | ret. | 0.5796 | 0.6720 | 0.9591 | 0.7959 | 0.8068 |
| | ours | 0.6019 | 0.6983 | 0.9606 | 0.7985 | 0.8055 |
| | CR | 0.6012 | 0.6351 | 0.9534 | 0.9556 | 0.8045 |
| CIFAR-100 | ret. | 0.6750 | 0.7048 | 0.7840 | 0.4430 | 0.4708 |
| | ours | 0.6072 | 0.6539 | 0.7863 | 0.5537 | 0.4712 |
| | CR | 0.5250 | 0.5922 | 0.7832 | 0.7267 | 0.4744 |

Due to page limitation, we have included the model performance for the class removal scenario in Appendix B. This additional data does not affect the conclusions above.

### 4.3 Case study: SISA

We apply FUNU to SISA to demonstrate its capability to reduce unlearning time while preserving model privacy. SISA [2] is a representative machine unlearning method. It initially splits the datasets into different shards and then divides the data in each shard into slices. Next, it trains a sub-model on each shard and sequentially adds training data from each slice. Finally, it aggregates the results from all sub-models to obtain the final model output.

In this configuration, when data removal requests are received, SISA first identifies the influenced shards and slices, i.e., the shards and slices containing the removal requests. It then retrains sub-models previously trained on these influenced shards and slices instead of retraining all sub-models, thereby reducing the time of unlearning.

*4.3.1 Experiment setting.* For the implementation of SISA, we set the number of shards to be five and the number of slices in each shard to be ten. In this case, we would have five sub-models and fifty data slices in total.

When applying FUNU to SISA, we first use the model which has the same structure as the sub-model and train it on the complete dataset for one epoch to get the reference model, as described in Section 3.2. For the unlearning scenario, We randomly choose 10, 30, and 50 samples as the removal requests $D_u$.

*4.3.2 Experiment results.* We analyze the experiment results from the perspectives of time cost and model privacy. In Figure 6 and Table 5, we refer to the methods "retrain" as unlearning the exact data removal requests, and "ours" indicates first filtering removal requests with FUNU.

**Time cost.** The comparison of time cost between using FUNU or not in SISA is shown in Figure 6. With FUNU, the average timing of applying SISA is reduced by 24%.

The number of influenced slices in SISA, denoted as $N_{IS}$, is shown in Table 5. The fewer slices are influenced, the less timing is for unlearning. With FUNU, the data removal requests are pruned, thereby reducing the number of influenced slices. The average decrease of $N_{IS}$ when applying FUNU is 31%, which aligns with the average decreased time cost.

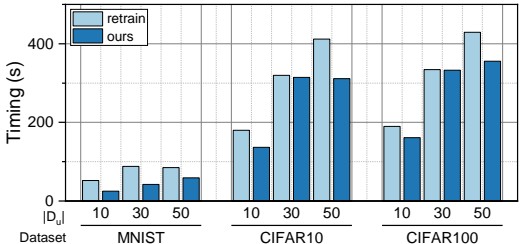

**Figure 6: Timing of applying FUNU to SISA**

**Model privacy.** Table 5 presents accuracy of both models on $D_t$, $D_u$, and $D_r$. On $D_t$ and $D_r$, the difference between accuracy for the retrained model and our model is less than 2%. On $D_u$, there is at most one sample where the two models predict differently. As the performance is similar in numerical, we thereby conclude that our method could lead to a model similar to the retrained model, consequently preserving model privacy.

## 5 Related work

Related research fields to this work are machine unlearning and data prototype discovery.

**Machine unlearning.** Machine unlearning explores how to obtain a model that closely resembles the retrained model. Since it has been proposed in [6], numerous studies have emerged in this area. As highlighted in the survey [44], mainstream research focuses on unlearning methods and verification mechanisms. Unlearning methods are categorized into two primary types: data reorganization [2, 6, 11, 15, 20] and model manipulation [17, 19, 21, 36]. The time cost of methods in the former category is more closely related to the size of removal requests. These methods typically manipulate the dataset based on the removal requests, and when there are fewer samples to unlearn, less manipulation is required.

**Data prototype discovery.** Data prototype discovery aims to identify prototypes that best represent the entire dataset. These prototypes can be used to reduce the training dataset size, thereby accelerating model training [24, 25], or to select potentially mislabeled data [33]. As discussed in Section 2.3, such methods have limitations in terms of efficiency, practicality, and adaptability to different types of removal requests, which prevent them from fully addressing our problem.

## 6 Discussion and conclusion

For this work, we have further discussion from the following two perspectives, including the potential impact of our method and the findings complementary to our research.

**Can FUNU benefit model-shifting unlearning methods?** Model-shifting methods [17, 19, 21, 36] constitute a group of unlearning techniques that directly update model parameters and introduce noise to achieve unlearning. For these methods, the efficiency of these methods is generally not significantly affected by the number of removal requests. However, these methods would have to consider deletion capacity, which is the maximum number of samples a model can unlearn while maintaining the privacy guarantee [36]. FUNU can be beneficial when the number of requests exceeds the deletion capacity. In such cases, FUNU could reduce the

**Table 4: Model performance of existing solutions under random removal**

| Dataset | Method | Parameter | $P^-$ | F1 of MIA | Accuracy of MIA | Accuracy on $D_r$ | Accuracy on $D_u$ | Accuracy on $D_t$ |
|---|---|---|---|---|---|---|---|---|
| MNIST | clustering | 0.2540 | 0.9100 | 0.5328 | 0.4358 | 0.9570 | 0.9656 | 0.9602 |
| | | 0.1000 | 0.7433 | 0.5078 | 0.4404 | 0.9569 | 0.9722 | 0.9604 |
| | | 0.0010 | 0.3022 | 0.5294 | 0.4487 | 0.9572 | 0.9689 | 0.9604 |
| | confidence | 0.1950 | 0.9411 | 0.5361 | 0.4312 | 0.9564 | 0.9900 | 0.9607 |
| | | 0.0620 | 0.8578 | 0.5189 | 0.3666 | 0.9566 | 0.9689 | 0.9602 |
| | | 0.0010 | 0.2378 | 0.5378 | 0.3401 | 0.9566 | 0.9867 | 0.9598 |
| | curvature | 0.5400 | 0.9656 | 0.5033 | 0.4562 | 0.9568 | 0.9656 | 0.9606 |
| | | 0.4990 | 0.7722 | 0.4822 | 0.3493 | 0.9569 | 0.9833 | 0.9597 |
| | | 0.4580 | 0.0767 | 0.5183 | 0.3598 | 0.9567 | 0.9756 | 0.9602 |
| CIFAR10 | clustering | 0.2990 | 0.8167 | 0.5900 | 0.6748 | 0.9542 | 0.8144 | 0.8021 |
| | | 0.1230 | 0.6722 | 0.6000 | 0.6914 | 0.9481 | 0.7978 | 0.7954 |
| | | 0.0010 | 0.4078 | 0.5439 | 0.6173 | 0.9598 | 0.8578 | 0.8069 |
| | confidence | 0.0750 | 0.9100 | 0.5722 | 0.5950 | 0.9504 | 0.7978 | 0.8067 |
| | | 0.0140 | 0.8200 | 0.5539 | 0.6285 | 0.9636 | 0.8733 | 0.8096 |
| | | 0.0010 | 0.5256 | 0.5206 | 0.6091 | 0.9534 | 0.8956 | 0.7988 |
| | curvature | 0.6660 | 0.9556 | 0.5911 | 0.6802 | 0.9617 | 0.8044 | 0.8095 |
| | | 0.6520 | 0.7344 | 0.5917 | 0.6871 | 0.9579 | 0.8167 | 0.8090 |
| | | 0.6380 | 0.0133 | 0.5356 | 0.6652 | 0.9526 | 0.9157 | 0.7996 |
| CIFAR100 | clustering | 0.2890 | 0.8756 | 0.6383 | 0.6740 | 0.7834 | 0.5033 | 0.4755 |
| | | 0.1180 | 0.7367 | 0.5956 | 0.6455 | 0.7840 | 0.5589 | 0.4709 |
| | | 0.0010 | 0.4633 | 0.5767 | 0.6471 | 0.7893 | 0.6222 | 0.5796 |
| | confidence | 0.0160 | 1.0000 | 0.6789 | 0.7103 | 0.7826 | 0.4467 | 0.4705 |
| | | 0.0010 | 0.9067 | 0.6061 | 0.6389 | 0.7874 | 0.4944 | 0.4728 |
| | curvature | 0.4060 | 0.9600 | 0.6932 | 0.7122 | 0.7833 | 0.4600 | 0.4710 |
| | | 0.3780 | 0.6689 | 0.6250 | 0.6768 | 0.7872 | 0.5533 | 0.4733 |
| | | 0.3510 | 0.0478 | 0.4883 | 0.6039 | 0.7873 | 0.8067 | 0.4737 |

\*CIFAR-100 only has two parameters when applying the confidence method because in this case
$max(avg(s) - std(s), 1e - 3) = avg(s)$.

**Table 5: Comparison between w/wo applying FUNU to SISA**

| Dataset | $|D_u|$ | Methods | $N_{IS}$ | Acc. on $D_t$ | Acc. on $D_u$ | Acc. on $D_r$ |
|---|---|---|---|---|---|---|
| MNIST | 10 | retrain | 21 | 0.9557 | 0.8000 | 0.9482 |
| | | ours | 9 | 0.9523 | 0.8000 | 0.9434 |
| | 30 | retrain | 38 | 0.9595 | 0.9333 | 0.9530 |
| | | ours | 17 | 0.9543 | 0.9333 | 0.9462 |
| | 50 | retrain | 40 | 0.9588 | 0.9600 | 0.9529 |
| | | ours | 25 | 0.9569 | 0.9400 | 0.9496 |
| CIFAR10 | 10 | retrain | 17 | 0.5062 | 0.6000 | 0.5759 |
| | | ours | 13 | 0.5034 | 0.6000 | 0.5740 |
| | 30 | retrain | 33 | 0.5114 | 0.4333 | 0.5833 |
| | | ours | 24 | 0.5099 | 0.4667 | 0.5799 |
| | 50 | retrain | 47 | 0.509 | 0.4000 | 0.5869 |
| | | ours | 30 | 0.5101 | 0.5000 | 0.5840 |
| CIFAR100 | 10 | retrain | 17 | 0.1075 | 0.1000 | 0.1575 |
| | | ours | 14 | 0.1061 | 0.1000 | 0.1604 |
| | 30 | retrain | 33 | 0.1168 | 0.1000 | 0.1721 |
| | | ours | 33 | 0.1164 | 0.1000 | 0.1724 |
| | 50 | retrain | 47 | 0.1154 | 0.1400 | 0.1753 |
| | | ours | 36 | 0.1148 | 0.1200 | 0.1724 |

size of the requests, thereby ensuring that the unlearning process remains protected by the privacy guarantee.

**Failure in verification vs. necessity to unlearn?** Recent work [46] has highlighted the fragility of current machine unlearning verification methods in some unlearning techniques. Specifically, it has shown that even with some data points not being removed, the model can remain indistinguishable from one that has undergone proper unlearning. This work can be seen as complementary to ours. Some removal requests would not significantly influence the retrained model, making it acceptable not to remove them. Our work primarily focuses on identifying such data points.

In conclusion, we first define unnecessary unlearning, review existing solutions, and then propose the FUNU method, which enhances the efficiency of machine unlearning methods while preserving model privacy. FUNU aim to filter out samples in data removal requests that would probably not lead to a model distinguishable from the retrained model. Theoretically, we prove its privacy guarantee. Empirically, we demonstrated that FUNU outperforms existing solutions in the balance of time cost, adaptability to different unlearning scenarios, and model privacy. We hope that our research can contribute to the machine unlearning field and better protect individuals' right to be forgotten in the big data era.

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

## A  Proof for Theorem 1.1

THEOREM A.1. *Suppose that for model $M_r$ and $M_u$, the logarithms of final FC layer output, denoted as $log(M_r)$ and $log(M_u)$, are $\lambda_1$-Lipschitz and $\lambda_2$-Lipschitz, and that $\|log(M_r(x)) - log(M_u(x))\| \leq \delta$ on $D_r$, then*

$$KL_{D_u^+}(p_u \| p_r) \leq n[(\lambda_1 + \lambda_2)(\sqrt{2 - 2\theta}) + \delta]$$

*where $p_u$ is the output distribution of $M_u$ on $D_u^+$, and $p_r$ is that of $M_r$. $n$ is the size of $D_u^+$.*

PROOF.  According to the definition of KL-Divergence,

$$KL_{D_u^+}(p_u \| p_r) = \mathbb{E}_{D_u^+}[p_u log(\frac{p_u}{p_r})]$$

As for $\forall x \in D_u^+$, there are more than $\alpha$ samples in $D_r$ that are similar to $x$. We denote it as a mapping: $nei(x_i) = x_j$, where $x_i \in D_u^+$ and $x_j$ is one of the samples similar to it in $D_r$. $nei(\cdot)$ is an injection function. Here we calculate the bound when there is only one sample in $D_r$ similar to $x$. As $\alpha \geq 1$, the bound we derived in the end is for the worst case.

The output distribution of $M_r$ on $nei(D_u^+)$ is denoted as $p_r^{sim}$, then we have

$$KL_{D_u^+}(p_u \| p_r) = \mathbb{E}_{D_u^+}[p_u log(\frac{p_u}{p_r})] \tag{5}$$

$$= \mathbb{E}_{D_u^+}[p_u log(\frac{p_u}{p_r^{sim}} \cdot \frac{p_r^{sim}}{p_r})] \tag{6}$$

$$= \mathbb{E}_{D_u^+}[p_u log(\frac{p_u}{p_r^{sim}}) + p_u log(\frac{p_r^{sim}}{p_r})] \tag{7}$$

$$= \mathbb{E}_{D_u^+}[p_u log(\frac{p_u}{p_r^{sim}})] + \mathbb{E}_{D_u^+}[p_u log(\frac{p_r^{sim}}{p_r})] \tag{8}$$

We first bound the second term. For each pair $x$ and $nei(x)$, where $x \in D_u^+$ and $nei(x) \in D_r$ ), as $cos(x, nei(x)) \geq \theta$, thus $\|x - nei(x)\| \leq \sqrt{2 - 2\theta}$. Due to that $log(M_r)$ satisfies $\lambda_1$- Lipschitz, $\|log(M_r(x)) - log(M_r(nei(x)))\| \leq \lambda_1 \|x - nei(x)\|_2$. Model outputs the prediction probability of sample $x$, consequently $\|M_u(x)\| \leq 1$. Combine these pieces together, we have

$$\mathbb{E}_{D_u^+}[p_u log(\frac{p_r^{sim}}{p_r})] = \mathbb{E}_{D_u^+}[p_u(log(p_r^{sim}) - log(p_r))] \tag{9}$$

$$= \sum_{x \in D_u^+} M_u(x)(log(M_r(nei(x))) - log(M_r(x))) \tag{10}$$

$$\leq \sum_{x \in D_u^+} (log(M_r(nei(x))) - log(M_r(x))) \tag{11}$$

$$\leq n\lambda_1 \|nei(x) - x\| \tag{12}$$

$$\leq n\lambda_1 \sqrt{2 - 2\theta} \tag{13}$$

$$\tag{14}$$

where $n$ is the size of $D_u^+$.

As for the first term, we denote the output distribution of $M_u$ on $nei(D_u^+)$ as $p_u^{sim}$, then we have

$$\mathbb{E}_{D_u^+}[p_u log(\frac{p_u}{p_r^{sim}})] \tag{15}$$

$$= \mathbb{E}_{D_u^+}[p_u log(\frac{p_u}{p_u^{sim}} \cdot \frac{p_u^{sim}}{p_r^{sim}})] \tag{16}$$

$$= \mathbb{E}_{D_u^+}[p_u log(\frac{p_u}{p_u^{sim}})] + \mathbb{E}_{D_u^+}[p_u(log(p_u^{sim}) - log(p_r^{sim}))] \tag{17}$$

Following in the same method in estimating $\mathbb{E}_{D_u^+}[p_u log(\frac{p_r^{sim}}{p_r})]$, we have $\mathbb{E}_{D_u^+}[p_u log(\frac{p_u}{p_u^{sim}})] \leq n\lambda_2 \sqrt{2 - 2\theta}$.

For $\mathbb{E}_{D_u^+}[p_u(log(p_u^{sim}) - log(p_r^{sim}))]$, as $\|log(M_r(x)) - log(M_u(x))\| \leq \delta$, thereby

$$\mathbb{E}_{D_u^+}[p_u(log(p_u^{sim}) - log(p_r^{sim}))] \tag{18}$$

$$= \sum_{x \in D_u^+} M_u(x)(log(M_u(nei(x))) - log(M_r(nei(x))) \tag{19}$$

$$\leq \sum_{x \in D_u^+} (log(M_u(nei(x))) - log(M_r(nei(x))) \tag{20}$$

$$\leq n\|log(p_u^{sim}) - log(p_r^{sim})\| \tag{21}$$

$$= n\delta \tag{22}$$

Consequently,

$$\mathbb{E}_{D_u^+}[p_u log(\frac{p_u}{p_r^{sim}})] \leq n\lambda_2 \sqrt{2 - 2\theta} + n\delta \tag{23}$$

With Eq. (14) and Eq. (23), we have

$$KL_{D_u^+}(p_u \| p_r) \leq n[(\lambda_1 + \lambda_2)(\sqrt{2 - 2\theta}) + \delta]$$

□

## B  Experiment supplymantary results

We present performance similarity between the model $M_u$ produced with our method FUNU and the retrained model $M_r$ under class removal scenario in Table 6. To showcase the similarity, we also use another machine unlearning method CR [21] as a baseline.

Except for the underlined values, in other metrics and datasets, our method is closer to the retrained model compared to the baseline. The mean accuracy difference between FUNU and $M_r$ over the three datasets was 0.0151, while the CR was 0.0708. The mean difference between FUNU's corresponding F1 of MIA and $M_r$ was 0.0686, while the CR was 0.0847. Thus the model generated by FUNU is more similar to $M_r$.

The performance of models generated using existing solutions is shown in Table 7, along with the parameters used. For existing solutions, their performance and $P^-$ are sensitive to thresholds. For the method "curvature", it corresponds to an average rate of change between $P^-$ and a threshold of 19.4723, i.e., a 0.01 decrease in the threshold will decrease P- by about 20%. While the average rate of change for confidence is 4.3357 and for clustering is 1.5018. This instability with thresholds is consistent with our conclusion in Section 4.2.4.

**Table 6: Model performance under class removal**

| Dataset | Method | Acc. of MIA | F1 of MIA | Acc. on $D_r$ | Acc. on $D_u$ | Acc. on $D_t$ |
|---------|--------|-------------|-----------|---------------|---------------|---------------|
| MNIST | retrain | 0.5904 | 0.5250 | 0.9561 | 0.9751 | 0.9605 |
| | ours | 0.5596 | 0.3911 | 0.9559 | 0.9704 | 0.9602 |
| | CR | 0.6594 | 0.6632 | 0.9506 | 0.8923 | 0.9508 |
| CIFAR10 | retrain | 0.6227 | 0.6542 | 0.9623 | 0.7013 | 0.8021 |
| | ours | 0.5804 | 0.6783 | 0.9532 | 0.7526 | 0.7948 |
| | CR | 0.4562 | 0.5803 | 0.9508 | 0.9942 | 0.8042 |
| CIFAR100 | retrain | 0.5767 | 0.5906 | 0.7817 | 0.5787 | 0.5143 |
| | ours | 0.5568 | 0.6384 | 0.7813 | 0.5568 | 0.4736 |
| | CR | 0.5376 | 0.6327 | 0.7832 | 0.7700 | 0.4744 |

**Table 7: Model performance of existing solutions under class removal**

| Dataset | Method | Parameter | $P^-$ | F1 of MIA | Accuracy of MIA | Accuracy on $D_r$ | Accuracy on $D_u$ | Accuracy on $D_t$ |
|---------|--------|-----------|-------|-----------|-----------------|-------------------|-------------------|-------------------|
| MNIST | clustering | 0.2540 | 0.8370 | 0.5660 | 0.3834 | 0.9555 | 0.9764 | 0.9598 |
| | | 0.1000 | 0.6780 | 0.5847 | 0.5787 | 0.9559 | 0.9763 | 0.9599 |
| | | 0.0010 | 0.4030 | 0.5609 | 0.3579 | 0.9561 | 0.9782 | 0.9593 |
| | confidence | 0.1950 | 0.9400 | 0.6216 | 0.5774 | 0.9558 | 0.9820 | 0.9601 |
| | | 0.0620 | 0.8950 | 0.5916 | 0.5315 | 0.9555 | 0.9809 | 0.9609 |
| | | 0.0010 | 0.5340 | 0.4024 | 0.5808 | 0.9563 | 0.9820 | 0.9600 |
| | curvature | 0.5400 | 0.9470 | 0.5660 | 0.4932 | 0.9561 | 0.9969 | 0.9607 |
| | | 0.4990 | 0.7240 | 0.5682 | 0.4613 | 0.9555 | 0.9813 | 0.9602 |
| | | 0.4580 | 0.0010 | 0.6282 | 0.6079 | 0.9573 | 0.9820 | 0.9603 |
| CIFAR10 | clustering | 0.2990 | 0.8367 | 0.5457 | 0.6411 | 0.9647 | 0.8038 | 0.8105 |
| | | 0.1230 | 0.6849 | 0.5559 | 0.6694 | 0.9344 | 0.8136 | 0.7851 |
| | | 0.0010 | 0.4008 | 0.4434 | 0.5561 | 0.9449 | 0.9263 | 0.7957 |
| | confidence | 0.0750 | 0.9379 | 0.6144 | 0.6346 | 0.9652 | 0.8202 | 0.8095 |
| | | 0.0140 | 0.8509 | 0.5788 | 0.6829 | 0.9640 | 0.7501 | 0.8052 |
| | | 0.0010 | 0.5313 | 0.4896 | 0.6387 | 0.9658 | 0.8676 | 0.8076 |
| | curvature | 0.6660 | 0.9179 | 0.5786 | 0.6967 | 0.9672 | 0.8114 | 0.8059 |
| | | 0.6520 | 0.5748 | 0.4581 | 0.5858 | 0.9665 | 0.9392 | 0.8087 |
| | | 0.6380 | 0.0293 | 0.5282 | 0.6645 | 0.9515 | 0.9081 | 0.7972 |
| CIFAR100 | clustering | 0.2890 | 0.8496 | 0.4469 | 0.5438 | 0.7803 | 0.4159 | 0.4738 |
| | | 0.1180 | 0.6726 | 0.5044 | 0.6164 | 0.7863 | 0.6106 | 0.4791 |
| | | 0.0010 | 0.3451 | 0.5797 | 0.6374 | 0.7827 | 0.6681 | 0.4670 |
| | confidence | 0.0160 | 0.9956 | 0.4624 | 0.5888 | 0.7901 | 0.3805 | 0.4646 |
| | | 0.0010 | 0.9159 | 0.4735 | 0.5897 | 0.7827 | 0.5177 | 0.4693 |
| | curvature | 0.4060 | 0.9735 | 0.5664 | 0.6475 | 0.7837 | 0.4071 | 0.4728 |
| | | 0.3780 | 0.6858 | 0.5398 | 0.6376 | 0.7801 | 0.5354 | 0.4747 |
| | | 0.3510 | 0.0796 | 0.5774 | 0.6348 | 0.7758 | 0.7788 | 0.4699 |

*CIFAR-100 only have two parameters when applying the confidence method because in this case $max(avg(s) - std(s), 1e - 3) = avg(s)$.