# OpenReview forum: "FUNU: Boosting machine unlearning efficiency by filtering unnecessary unlearning"
_ACM.org/TheWebConf/2025/Conference — WWW 2025 Poster_

### Official Review · Reviewer_bFUa · 2024-11-15

**Novelty:** 6
**Technical Quality:** 5

**Review:**

# Primary Concern: Irrelevance to Security and Privacy
The FUNU method is essentially an optimization for computational efficiency in machine unlearning and has no direct relation to security and privacy. Its "privacy guarantee" only refers to consistency in model behavior after unlearning and does not provide actual data privacy protection or security measures (such as data encryption, access control, or prevention of data leakage). Therefore, FUNU does not meet the focus of this track on security and privacy.


# Quality
The experimental design is sound, clearly demonstrating FUNU’s improvements in time efficiency with effective comparisons to baseline methods like SISA. However, this work has no direct relevance to the field of security and privacy.

# Clarity
The paper is well-structured and logically presented. However, the term "privacy guarantee" may mislead readers into thinking that this method is related to actual data privacy protection. It is recommended to clearly distinguish between FUNU’s computational consistency and genuine privacy protection.

# Originality
The paper introduces a novel "filtering unnecessary unlearning" approach, improving unlearning efficiency by identifying similar data points. This is innovative as a computational optimization but is unrelated to the fields of privacy and security.

# Significance
FUNU has practical value in unlearning scenarios that require frequent data removal. However, it does not directly contribute to the field of Web security and privacy, making it unsuitable for this track.

# Pros
In this paper, they propose a new concept that can filter unnecessary unlearning requests. This method can take a reader some inspiration.

# Cons
1. **The core problem:** FUNU is fundamentally an optimization for unlearning efficiency and does not involve data privacy or security protection, **failing to meet the security and privacy track’s requirements!!!!!**

2. Small problem: The formatting of the tables and images needs a bit more work

**Questions:**

Your job and security have nothing to do with the privacy track. If I understand correctly, by privacy, you mean that unlearning can remove private data? Or that after the data is removed, the model does not show any trace of ever containing the data?

**Reviewer Confidence:**

3: The reviewer is confident but not certain that the evaluation is correct

**Scope:**

3: The work is somewhat relevant to the Web and to the track, and is of narrow interest to a sub-community

---

### Official Review · Reviewer_F3pg · 2024-11-29

**Novelty:** 6
**Technical Quality:** 5

**Review:**

The paper introduces FUNU, a novel approach aimed at improving the efficiency of machine unlearning by identifying and bypassing "unnecessary unlearning" requests. It is technically robust, with well-defined methodologies and thorough experimentation.

Pros:

1.Innovatively tackles the inefficiency of traditional machine unlearning methods.

2.Strong experimental results demonstrating significant reductions in unlearning time.

3.Provides theoretical privacy guarantees alongside practical benefits.

Cons:

1.This paper is not closely related to the web.

2.Only one unlearning method has been proven to be effective for FUNU.

**Questions:**

1.Why use an example of a recommendation system in the introduction, but conduct experiments on image datasets?

2.No pseudocode provided for the algorithm.

3.How effectively can FUNU integrate with methods other than SISA, such as model manipulation approaches?

**Reviewer Confidence:**

2: The reviewer is willing to defend the evaluation, but it is likely that the reviewer did not understand parts of the paper

**Scope:**

2: The connection to the Web is incidental, e.g., use of Web data or API

---

### Official Review · Reviewer_xtCF · 2024-12-02

**Novelty:** 5
**Technical Quality:** 5

**Review:**

FUNU introduces a practical, efficient, and adaptive framework for machine unlearning by leveraging the concept of unnecessary unlearning. It minimizes computational costs while meeting privacy requirements, making it a promising solution for real-world applications in privacy-preserving machine learning.

**Questions:**

Strengths:

The introduction of "unnecessary unlearning" to reduce the size of data removal requests based
on the fundamental assumption that the removal of certain data would not result in a distinguishable retrained model is practical.

It improves efficiency by improving the number of data points to be removed.

Formulation of security notions is given.

FUNU adapts to diverse unlearning scenarios, such as random removal and class removal,



Weaknesses:

FUNU requires considerable computation time to find the similar data points, computing similarity matrix and finding the cosine angle threshold

With unnecessary unlearning, the approach does not remove some data if similar data points are found in the existing data-points. But, there is a chance some crucial data-points will not be removed because of this situation.

Suppose there are three similar points, A, B, and C. The request came for A, but according to the algorithm, A will not be removed. Now, the request came for B, it will also not be removed. Now, the request came for C. What will happen now? Can you clarify?

The method's performance still depends on accurately tuning parameters like similarity thresholds, which could be dataset-specific and can easily be miscalculated.

There is no reference literature provided on “In this case, we infer the samples correctly predicted by 𝑀_ 𝑟𝑒𝑓 have sufficient similar neighbors in the entire dataset” on section 3.2

Question: Why do authors choose to compute the distance matrix based on the feature generated, not comparing the raw dataset itself.

Question: “In this case, we infer the samples correctly predicted by 𝑀_ 𝑟𝑒𝑓 have sufficient similar neighbors in the entire dataset”, is there any literature to support this claim.

**Reviewer Confidence:**

2: The reviewer is willing to defend the evaluation, but it is likely that the reviewer did not understand parts of the paper

**Scope:**

3: The work is somewhat relevant to the Web and to the track, and is of narrow interest to a sub-community